# Implicit Warping for Animation with Image Sets

**Arun Mallya**
amallya@nvidia.com

**Ting-Chun Wang**
tingchunw@nvidia.com
NVIDIA

**Ming-Yu Liu**
mingyul@nvidia.com

## Abstract

We present a new *implicit warping* framework for image animation using sets of source images through the transfer of the motion of a driving video. A single cross-modal attention layer is used to *find correspondences* between the source images and the driving image, *choose* the most appropriate features from different source images, and *warp* the selected features. This is in contrast to the existing methods that use explicit flow-based warping, which is designed for animation using a single source and does not extend well to multiple sources. The *pick-and-choose* capability of our framework helps it achieve state-of-the-art results on multiple datasets for image animation using both single and multiple source images.

## 1 Introduction

We study the task of generating videos by animating a set of source images of the same subject using the motions of a driving video possibly containing a different subject. This is more general than the setting of animating a single source image studied in the prior works [26, 27, 41, 50] as it is a special case when the set contains just one image. Moreover, having the capability of animating a set of images, especially those representing different views when available, is of great practical value. A single image often cannot fully describe the subject due to occlusions, limited pose information, *etc*. Diverse source images provide more appearance information and reduce the burden of hallucination that an image generator has to perform. As shown in Fig. 1, multiple source images provide more complete information, such as the color of the eyes, the texture of the background, *etc*. This allows for potentially generating an output image that is more faithful to the source setting.

The single-source-based prior works [26, 27, 41, 50] ubiquitously rely on explicit flow-based warping of the source image conditional on the pose of the driving image. Due to this architectural choice, they often have to be modified in ad-hoc ways to take advantage of multiple source images. One scheme is to train an additional pre-processing network to select the most appropriate source image for the given driving image. This would, however, not allow for the use of features from multiple source images at a time. The other possibility is to warp each source image to the driving pose and then average the now-aligned warped features for the generator input. But as is visible in Fig. 1 and later measured in Section 4, this leads to sub-optimal results due to the misalignment of warped features and inconsistent predictions across views.

In this work, we introduce a novel *implicit warping* mechanism that overcomes the above drawbacks—(1) It operates on a set of images and naturally scales from a single source image to multiple source images; and (2) Without predicting explicit per-source flow, a single layer finds correspondences between the source images and the driving image, chooses the aptest features from different source images, and performs warping of the selected features. Our novel layer and the ability to *pick-and-choose* appropriate features from a set of source images allows our method to achieve state-of-the-art image animation quality in the multiple source image setting, as well as the single source image setting. The benefits of our framework are obvious in Fig. 1, in the image reconstruction and cross-identity motion transfer tasks. In the top row, the inability of prior works of FOMM [26] and face-vid2vid [41] to pick-and-choose features results in an incorrect output eye color. Only our

36th Conference on Neural Information Processing Systems (NeurIPS 2022).

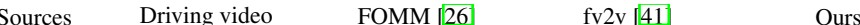

| Sources | Driving video | FOMM [26] | fv2v [41] | Ours |
|---------|---------------|-----------|-----------|------|

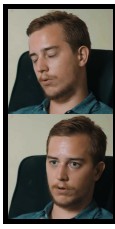

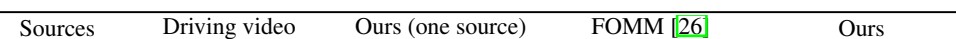

| Sources | Driving video | Ours (one source) | FOMM [26] | Ours |
|---------|---------------|-------------------|-----------|------|

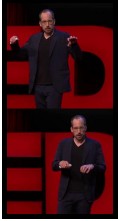

Fig. 1: Our image animation model uses information from one or more source images by picking and choosing from available features. This allows our model to output for *e.g.*, the correct eye color, and the correct background as highlighted by the orange ovals, in both same- and cross-identity motion transfer. Prior works fail to correctly align and utilize multiple source images, leading to blurry and incorrect outputs. Click on any result to play the video. *Please view with Adobe Acrobat Reader.*

method produces the correct background appearance in the lower row when given both source images. The orange ovals highlight areas of major differences amongst predicted outputs.

## 2 Related Work

**Image animation.** Recent works on image animation can be roughly categorized into subject-dependent and subject-agnostic works. For subject-dependent, the framework is trained on a specific subject and can only animate that specific person [2, 8, 30–33, 36, 46]. For example, NerFACE [8] trains a neural radiance field to model a target face using a short clip of the subject. On the other hand, subject-agnostic frameworks only need a single 2D image to perform the animation on the target person, so they are usually more general and applicable [1, 3, 5, 6, 9–11, 16, 21, 23, 26, 27, 29, 37, 40, 41, 44, 50, 51, 54]. For example, Siarohin *et al.* [26] predict local affine transformations using learned latent keypoints. Zakharov *et al.* [50] focus on inference speed and propose to decompose high and low-frequency components for better efficiency. Wang *et al.* [41] learn 3D latent keypoints and their decomposition to predict warping flows and gain better controllability. Doukas *et al.* [6] adopt 3D morphable models to generate a dense flow field to warp the source image and then take the warped input along with audio features to synthesize the final output. While these frameworks achieve promising results in image animation, none of them are designed for using multiple reference source images as input. In contrast, our framework is designed to take advantage of complementary information available in different input images for high-quality image animation.

**Attention in computer vision.** The self-attention layer, which uses the same input as the query, key, and value, was popularized by the work of Vaswani *et al.* [35] for the task of machine translation. Since then, a number of works have explored using the self-attention layer in conjunction with convolutional layers for the tasks of image and video recognition [13, 14, 38, 39, 43, 45, 47]. A few works have also explored cross-modal attention in which the queries, keys, and values are computed from two different domains, such as vision and text [48, 49], audio and text [19], *etc*. In our cross-modal attention layer, the queries and keys are from the domain of a compact image representation, such as keypoints, while the values are extracted from the dense image features.

Networks fully composed of self-attention layers, also known as transformers, have recently obtained state-of-the-art numbers in the tasks of image classification [18, 20, 34, 42], as well as generation [7, 15, 17, 22, 24, 25]. Our generator uses a hybrid architecture, with a single cross-modal attention layer for warping the features of the source image conditional on the pose of the driving image. This is followed by a decoder made up of convolutional residual blocks to produce an output image.

While we do not predict explicit flows to warp feature maps, related work by Xu *et al.* [47] has used efficient 1-D attention layers for predicting explicit optical flow. We use global 2-D attention, but such factored 1-D attention layers and spatial-reduction attention layers [42] can be used to reduce the run-time further. To the best of our knowledge, we are the first to propose a way of finding correspondences and warping features via a single cross-modal attention layer for image animation.

## 3   Image Animation with Implicit Warping

The idea behind image animation using a source image is to warp features of the source image conditional on the given driving image so as to reconstruct the driving image or transfer the pose of the driving image in the case of cross-identity motion transfer. Such image animation methods typically consist of 4 components: (1) Source image representation extraction, (2) Driving image representation extraction, (3) Source image feature warping, and (4) Warped image feature decoding. In this work, we focus on the third component—source image feature warping. The other components in our method are largely based on prior works [26, 27, 41]. In the following sections, we describe our framework and contributions in more detail.

**Explicit flow-based warping.** Prior works on image animation, such as Monkey-Net [27], First Order Motion Model (FOMM) [26], and face-vid2vid [41], use a flow-based warping technique to deform the features extracted from the source image. They generate a warping field using the compact representations extracted from the source and driving images, usually in the form of keypoints and additional related information such as Jacobian matrices. This field is used to warp the source image features, which are subsequently mapped to the output image space by a decoder network. The prior works, specifically FOMM [26] and face-vid2vid [41], perform warping using the scheme illustrated in Fig. 2a. Given the locations of $K$ unsupervisedly-learned keypoints in the source image $S$ and driving image $D$, they first generate $K$ per-keypoint flows $w_1, \cdots, w_K$. By using the source image (or extracted features) warped with each of the $K$ flows, a *motion estimator* network predicts how to combine the individual flows in the *flow composition* mask $m$. Combining the $K$ per-keypoint flows with this mask gives the final *composite flow* mask $w$, which is applied to the source image features $f_S$ to obtain the warped feature $w(f_S)$. Due to the per-keypoint warping steps, the time complexity of these methods is directly proportional to the number of keypoints $K$.

As mentioned in Section 1, the prior works have significant drawbacks when extending them to multiple source images. We either have to average the per-source image warped features or employ a heuristic to choose the most appropriate source image given the driving image. Neither of these two solutions is ideal as the former can lead to blurry outputs and the latter leads to flickering when switching between sources. A more desirable way is to have an end-to-end trainable pick-and-mix feature combination mechanism that can aggregate information from multiple source images.

In this work, we introduce '*Implicit Warping*'—a new and simpler way to warp source features. This results in an image animation method that is easily and directly extensible to using multiple source images. Our method relies on a cross-modal attention layer illustrated in Fig. 2b. It neither produces explicit per-keypoint warping flows nor requires per-source image warping of features.

**Implicit Warping using attention.** The process of warping the source image features conditional on the driving image consists of two parts: (1) Identifying dense correspondences between the source

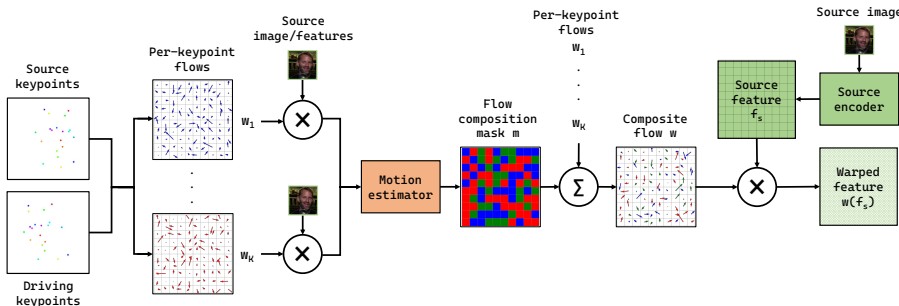

(a) Warping mechanism used in First Order Motion Model (FOMM) [26] and face-vid2vid [41].

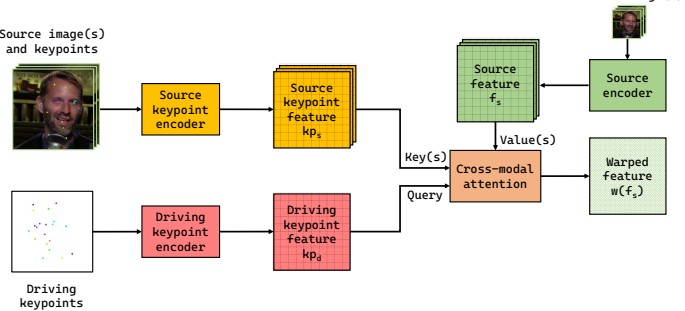

(b) Our warping mechanism based on cross-modal attention.

Fig. 2: Comparison of image warping mechanisms of the prior works against ours. Our method is based on cross-modal attention and is simpler. Moreover, it can be easily extended to using multiple source images. Our method does not require per-keypoint warping of the source image(s) or features unlike FOMM [26] and face-vid2vid [41], and can pick and mix features from multiple images.

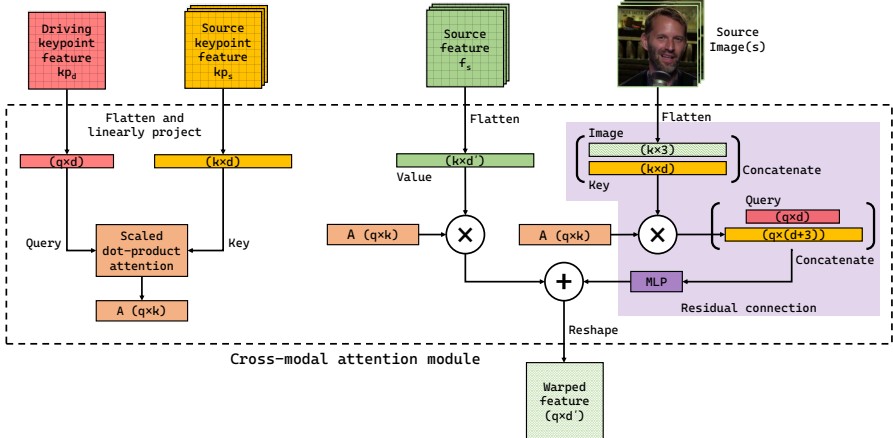

Fig. 3: Coss-modal attention used for warping. The features from the driving keypoints and source keypoints are treated as the *query* and the *key* respectively. The computed attention is applied on the source features, or the *value*. This reconfigures the source features to correspond to the pose of the driving image. The attention is also applied to the source image pixels and the *key*, which are then concatenated with the *query*. After processing with an MLP, these are added to the output values. Reshaping the outputs back to 2D produces the warped feature. This scheme can be naturally extended to multiple source images. Adding a new source image increases the number of keys and values, resulting in an appropriately larger attention map A (specifically dimension $k$ in above figure).

and driving images, and (2) Warping the source image features based on the dense correspondences. The key intuition behind our method is that these two steps can be represented using a single scaled dot-product attention layer popularized by Vaswani *et al.* [35],

$$\text{Attention}(Q, K, V) = \text{softmax}\left(\frac{Q \times K^T}{c}\right) \times V = S \times V, \tag{1}$$

where $Q$, $K$, and $V$ are made up of vector-valued queries, keys, and values, respectively, and $c$ is a scalar-valued scaling factor. For a given query vector $Q$ of size $(1 \times d)$, the softmaxed value of the product $Q \times K^T$ can be interpreted as the similarity $S$ of the given query vector with the set of all available keys $K$ of size $(k \times d)$. After the softmax normalization, the key with the highest dot-product with the query will obtain the highest similarity. After computing the similarity $S$ of size $(1 \times k)$ and given values $V$ of size $(k \times d')$, the product $S \times V$ gives a query-conditional weighted average of values of size $(1 \times d')$. This scheme naturally extends to multiple queries $Q$ of size $(q \times d)$. In our particular application, the queries $Q$ come from the driving keypoints, and the keys $K$ and values $V$ come from the source image(s) keypoints and features, respectively. Using additional source images only leads to an increase in the number of available keys and values.

We refer to the above procedure as *implicit warping*, as illustrated in Fig. 2b. This is opposed to explicit warping shown in Fig. 2a, where an explicit flow map is first computed from the source and driving keypoints and then applied to source features to produce warped outputs. There is no explicit flow map in our scheme; the similarity between the queries and keys produces an implicit flow map, which is then applied to the values to produce the warped outputs. In the following, we describe how we learn queries, keys, and values for the novel implicit warping based on cross-modal attention.

**Learning queries, keys, and values.** Similar to the prior works [26, 27, 41], we use learned keypoints and associated information as a compact representation of the source and driving images. Our image encoder that predicts keypoints reuses the architecture from FOMM [26]. While reconstructing the driving image, we only have access to the driving keypoints but have access to both the source image(s) and the source keypoints. Given the $K$ keypoints predicted from an image, we create a spatial representation of $K$ channels where each channel represents a keypoint. For the $k^{th}$ keypoint, we place a Gaussian of a fixed mean and variance in channel $k$, centered at the predicted keypoint location. In addition to the $K$ keypoints, we predict a scalar per-keypoint with values in $[0, 1]$, serving as the keypoint strength score. We multiply this predicted scalar with the channel corresponding to the keypoint. This allows us to modulate the keypoint *strength*, which is related to its visibility, in the given source and driving images. The predicted keypoint strengths are later discussed in Section 4 and visualized in Fig. 6. Both the keypoint locations and the scalar factors are learned in an unsupervised manner. Unlike FOMM [26], we do not predict the Jacobian matrix per keypoint in addition to the keypoint location, making our intermediate representation more compact ($2 + 1 = 3$ values per keypoint instead of $2 + 4 = 6$ values per keypoint in the prior work). Unlike face-vid2vid [41], we do not assume rigid motion and find applications beyond talking-head animation.

We use a U-net to encode the above spatial driving keypoint representations, with a final point-wise projection layer to map it to the dimension $d$ selected for the queries and the keys. A similar U-net is used to encode the source image, except that its input consists of both the source image and the spatial source keypoint representation. Both these networks accept inputs and produce outputs at $1/4$ resolution of the source image, *i.e.* 64 for $256 \times 256$, 96 for $384 \times 384$, *etc*. Assuming a source image size of $256 \times 256$, the two U-net networks produce a total of $(64 \times 64 \times d)$, or $(4096 \times d)$ *queries* and *keys*, for each driving and source image, respectively. Since the output of these U-nets is of a fixed dimension $d$, the design of the subsequent components is independent of the number of keypoints used. In order to obtain *values* from the source image, we pass it through two downsampling convolution layers and obtain values of size $(4096 \times d')$. Note that there is a one-to-one spatial correspondence between the keys and the values. Additional details can be found in the supplementary material.

However, there is one possible pitfall to the scheme described so far. If none of the keys have a high dot-product with the query, the key with the highest dot-product will still be assigned the high similarity after the softmax, ensuring that the outputs sum up to 1. While it is the key most similar to the query in the given set of keys, it may not be *similar enough* to produce a good output. For example, suppose the source image has a face with lips closed, while the driving image has one with lips open and teeth exposed. In this case, there will be no key (and value) in the source image appropriate for the mouth region of the driving image. We overcome this issue by allowing our method to learn additional image-independent key-value pairs, which can be used in the case of missing information in the source image. These additional keys and values are concatenated to the keys and values obtained from the source image. In our experiments, we add 400 extra keys and values, approximately 10% of the number of keys and values obtained from a single source image of size $256 \times 256$. We use semantic dropout while training our framework to encourage the learning and use of the added keys and values. This is achieved by randomly dropping out keys and values corresponding to different facial regions, such as eyebrows, eyes, lips, teeth, *etc*. This is possible as

an attention layer treats its input keys and values as a set. In contrast, convolutional layers assume a specific spatial order, and spatial locations cannot be randomly dropped out. As shown later in Section 4 and Table 4, these extra keys and values help improve output quality.

**The cross-modal attention module.** After obtaining queries, keys, and values, the cross-modal attention module computes the warped source features conditional on the driving image via implicit warping. As shown in Fig. 3, the queries $kp_d$ obtained from the driving image are of size $(q \times d)$ after flattening. The keys $kp_s$ and values $f_s$ obtained from the one or more source images are of size $(k \times d)$ and $(k \times d')$ respectively. We add learnable position embeddings to the queries $kp_d$ and keys $kp_s$. The dot product $Q \times K^T$ after scaling and softmax gives the attention matrix $A$ of size $(q \times k)$. By multiplying $A$ with values $V$ of size $(k \times d')$, we obtain the warped output feature of size $(q \times d')$.

Additionally, we concatenate the image pixels with the keys and align them with the driving image by multiplying them with the computed attention map, which is further concatenated with the queries and passed through an MLP (purple-shaded part of Fig. 3). The intuition behind using the warped keys and queries is to recover any information, such as skew or rotation, which a weighted average of values may not easily capture. Further, including the pixel values helped by aiding color consistency. Adding this residual connection helped improve the quality of outputs, as shown in Section 4 and Table 4.

While our end-to-end framework is reasonably fast ($\sim 10$ FPS on $512 \times 512$ images), it is possible to reduce the run-time of the attention operations by utilizing the factored 1-D attention layers [47] or the spatial-reduction attention (SRA) layer [42], which we leave for the future work. After training, we observed that the learned attention masks are very sparse. This indicates that we can further reduce the inference time by selecting just the top-few values per row from the product $A = Q \times K^T$, and multiplying them with corresponding values of $V$, thereby saving computation.

We show results on multiple datasets in the single and multiple source image settings with comparison to strong baselines in the next section, and try to provide insights into the workings of the model.

## 4 Experiments

**Datasets.** We perform our ablations on the TalkingHead-1KH [41] dataset at the $256 \times 256$ resolution and compare with baselines at the full $512 \times 512$ resolution. We also report results on the $256 \times 256$ VoxCeleb2 [26] dataset. Additionally, to demonstrate the generality of our method, we report results on the more challenging TED Talk [28] dataset of moving upper bodies at a resolution of $384 \times 384$.

**Metrics.** We measure the fidelity of driving image reconstruction using PSNR, $\mathcal{L}_1$, LPIPS [53], and FID [12]. The quality of motion transfer is measured using average keypoint distance (AKD) between keypoints predicted using MTCNN [52] for faces and OpenPose [4] for upper bodies. For upper-body motion transfer, we also report the missing keypoint ratio (MKR), at a chosen keypoint prediction confidence threshold $C$, together denoted as (AKD, MKR)@$C$. For human evaluation, MTurk users were shown videos synthesized by two different methods and asked to choose the more realistic one.

**Baselines.** On the face datasets, we compare our method against two state-of-the-art methods: First-Order Motion Model (FOMM) [26], and face-vid2vid (fv2v) [41]. On the upper-body dataset, we compare against FOMM [26] and the state-of-the-art AA-PCA [28]. We use the pretrained model provided for AA-PCA, while we train all other methods from scratch. As previously mentioned in Section 1, prior works are unable to handle multiple source images, despite the immense practicality of the setting. We adapt these methods, where possible, to multiple source images by warping features from each provided source image, and then averaging them to obtain a single warped feature map. All methods use $K = 20$ keypoints in the following experiments. Additional details about datasets, metrics, and training hyper-parameters for all methods are provided in the supplementary material.

**Driving image reconstruction with a single source image.** This is the standard evaluation setting used in prior work [26–28, 41]. The first frame of each video is chosen as a source image, and the full-length videos are reconstructed. Quantitative evaluation of predictions is presented in Table 1. Our method outperforms competing methods in almost all metrics on both datasets. We obtain a lower FID and average keypoint distance (AKD) on both datasets, indicating good visual quality as well as motion reproduction in the predictions. Fig. 4 highlights a case where our attention-based method is able to provide the correct reconstruction while other methods fail. As attention is non-local and global in nature, it can borrow image features from regions that are spatially far away, *e.g.* the left and right hair and ears. Prior works based on explicit flow prediction would have to predict large values of flow for such newly disoccluded regions. Fig. 5 compares results between the various methods

Table 1: Comparisons with prior work on image reconstruction using a single source image.
↑ larger is better, ↓ smaller is better. See section 4 for details about metrics.

| | TalkingHead-1KH [41] 512 × 512 faces | | | VoxCeleb2 [26] 256 × 256 faces | | | TED Talk [28] 384 × 384 upper bodies | | |
|---|---|---|---|---|---|---|---|---|---|
| | FOMM [26] | fv2v [41] | Ours | FOMM [26] | fv2v [41] | Ours | FOMM [26] | AA-PCA [28] | Ours |
| PSNR ↑ | 23.23 | 23.27 | **23.32** | **24.15** | 23.66 | 23.54 | 24.17 | 25.14 | **25.24** |
| $\mathcal{L}_1$ ↓ | 12.86 | **12.30** | 12.86 | 10.99 | **11.10** | 11.40 | 8.22 | **6.87** | 7.69 |
| LPIPS ↓ | 0.16 | 0.16 | **0.15** | 0.12 | 0.12 | **0.11** | 0.16 | 0.13 | **0.12** |
| AKD (MTCNN) ↓ | 4.25 | 3.83 | **3.48** | **1.69** | 1.70 | 1.75 | – | – | – |
| (AKD, MKR)@0.1 ↓ | – | – | – | – | – | – | (7.44, 0.01) | (5.10, 0.005) | **(4.39, 0.005)** |
| (AKD, MKR)@0.5 ↓ | – | – | – | – | – | – | (5.93, 0.06) | (4.13, 0.030) | **(3.31, 0.023)** |
| FID ↓ | 18.15 | 18.09 | **16.44** | 9.53 | 5.76 | **4.68** | 24.69 | 19.17 | **18.27** |

Table 2: Comparisons with prior work on image reconstruction using multiple source images.

| TalkingHead1KH (512 × 512 faces) | | | | | | | | | |
|---|---|---|---|---|---|---|---|---|---|
| #Source frames | 1 | | | 2 | | | 3 | | |
| | FOMM [26] | fv2v [41] | Ours | FOMM [26] | fv2v [41] | Ours | FOMM [26] | fv2v [41] | Ours |
| PSNR ↑ | 24.260 | 24.273 | **24.276** | 25.433 | 25.557 | **26.094** | 26.044 | 25.843 | **26.768** |
| $\mathcal{L}_1$ ↓ | 11.075 | **10.338** | 11.152 | 9.023 | 8.292 | **7.990** | 8.238 | 8.330 | **7.347** |
| LPIPS ↓ | 0.139 | 0.135 | **0.130** | 0.116 | 0.110 | **0.089** | 0.109 | 0.118 | **0.081** |
| AKD (MTCNN) ↓ | 5.039 | 4.792 | **4.370** | 4.662 | 4.384 | **4.029** | 4.483 | 6.201 | **3.957** |
| FID ↓ | 17.454 | 17.174 | **15.658** | 19.647 | 16.846 | **9.850** | 19.495 | 19.216 | **9.097** |

| TED Talk (384 × 384 upper bodies) | | | | | |
|---|---|---|---|---|---|
| #Source frames | 1 | | 2 | | 3 | |
| | FOMM [26] | Ours | FOMM [26] | Ours | FOMM [26] | Ours |
| PSNR ↑ | 24.096 | **25.188** | 24.546 | **26.528** | 24.835 | **26.884** |
| $\mathcal{L}_1$ ↓ | 8.290 | **7.737** | 7.726 | **6.597** | 7.398 | **6.319** |
| LPIPS ↓ | 0.156 | **0.119** | 0.143 | **0.088** | 0.137 | **0.081** |
| (AKD, MKR)@0.1 ↓ | (7.618, 0.010) | **(4.438, 0.005)** | (6.898, 0.010) | **(4.182, 0.004)** | (6.518 0.010) | **(3.989, 0.004)** |
| (AKD, MKR)@0.5 ↓ | (6.041, 0.060) | **(3.349, 0.025)** | (5.325, 0.067) | **(3.264, 0.021)** | (5.064, 0.067) | **(3.173, 0.021)** |
| FID ↓ | 24.324 | **17.746** | 29.053 | **13.822** | 30.838 | **13.128** |

when using a single source image on the TED Talk dataset. Our method demonstrates better hand and face reconstruction than prior works.

**Driving image reconstruction with multiple source images.** We now compare results in the case of driving image reconstruction when multiple source images are provided. This is a setting of great practical value as a single source image often does not have all the details necessary to reconstruct a driving image with any arbitrary pose and expression. We present results obtained when using multiple source images in Table 2. Here, we use sequences of length at most 180 frames and select regularly-spaced source images in addition to the first frame. Similar to the single source image setting, our method outperforms all prior work. As the number of source images increases, our method obtains better reconstructions as indicated by the improving scores on all metrics. However, reconstructions by prior work get worse as the number of source images increases, contrary to expectation. Upon viewing the qualitative results shown in Fig. 1, the issue with prior works becomes clear—even though features from each source image are warped to the same driving pose, there exist misalignments between the warped features. Further, they do not have the ability to choose features from only a specific source image. Our method does not suffer from this issue due to global attention over all available images and their features. As a result, our method is able to predict the correct eye color and sharp background, as seen in Fig. 1. Fig. 4 shows regions of the source image that the highlighted regions of the output borrow features from. This demonstrates our *picking-and-choosing* mechanism—regions $\{1, 2, 4, 7, 9\}$ visible in both sources contribute equally to the output, while the rest, only visible in one of the sources, is used exclusively. User preference scores are provided in Table 3. Users prefer our method in all settings, and the preference increases further when more source images are used. Additional comparisons are available in the supplementary material.

**A closer look at the architecture.** Having shown the superiority of our method on image reconstruction in the single and multi-source image settings, we now examine the contribution of architectural choices to the output quality. As shown in Table 4, adding the residual connection visualized by the purple region in Fig. 3, improves all metrics. This indicates that the warped image and keys, along with the queries provide additional information to the values. Adding extra learnable keys and values, as described in Section 3 further improves all metrics by allowing the network to share features when information is missing in the source image(s). Fig. 6 visualizes predicted keypoints and strengths for

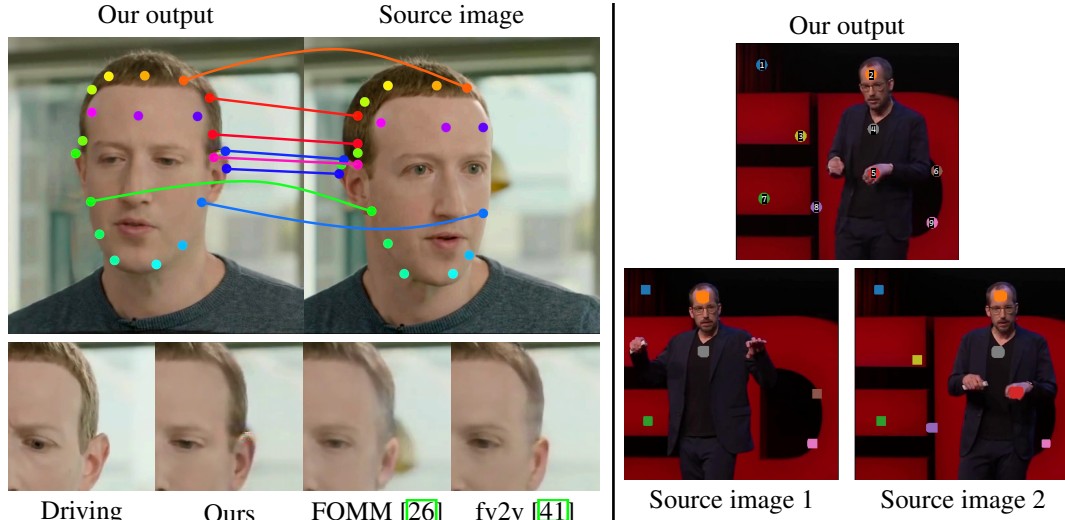

Fig. 4: Benefits of non-local attention. In the 2 top-left images, we visualize the locations in the source image assigned the highest attention score for correspondingly marked regions in our output. Note that the features for the newly disoccluded right hair and right ear of the output are borrowed from the left side of the source image. The 4 images on the bottom-left side show zoomed-in regions of the driving image, and reconstructions by 3 different methods. Due to global attention on the source, our method produces the correct hair color for the disoccluded regions, while the rest fail to do so. To generate the output image on the top-right, our method *picks-and-chooses* features from both or one of the source images on the bottom-right, depending on the visibilities of various features, *e.g.* region 1 is used from both images, while region 8 is only from source image 2.

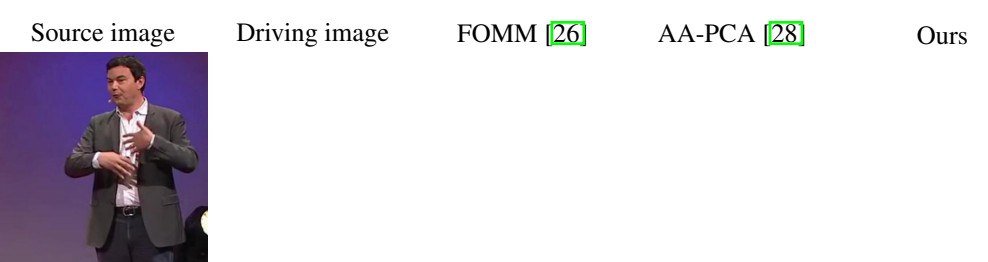

Fig. 5: Visual comparison of results on the TED Talk dataset. Our method produces better motion, face, and hand quality. Click on any result to play video. *Please view with Adobe Acrobat Reader.*

various configurations of a person's face. For the same arrangement of keypoints (both or single eye closed), we see different keypoint strengths based on the appearance. We also see large changes in keypoint strengths when certain keypoints are occluded, *e.g.* when the face is rotated to an extreme angle. This single scalar in place of a 4-valued Jacobian matrix per keypoint helps us achieve a more compact representation useful for applications such as video compression and conferencing [41].

## 5 Discussion

In this work, we presented the novel *implicit warping* framework and showed its application to image animation. Through the use of a single cross-modal attention layer, our method is able to deform source image features conditional on the driving image, and can scale to multiple source images as-is. Unlike prior work, our method does not predict explicit per-source image flows, and can pick-and-choose from multiple source image features. It convincingly beats prior state-of-the-art methods on multiple datasets in both the single and multiple source image settings, while using a more compact intermediate representation of keypoint locations and strengths.

**Limitations.** Our method can fail in cases where it is expected to hallucinate a lot of missing information. For example, given just a source image of the back of someone's head, it cannot generate

Table 3: User preference scores. Users prefer our method even when using one source, with increased preference as more sources are used.

| TalkingHead-1KH | vs FOMM | vs fv2v |
|---|---|---|
| 1 source | 64.1% | 53.5% |
| 2 sources | 65.8% | 59.9% |
| TED Talk | vs FOMM | vs AA-PCA |
| 1 source | 81.5% | 63.3% |
| 2 sources | 91.4% | – |

Table 4: Architecture ablations on image reconstruction with a single source image on TalkingHead-1KH at $256 \times 256$ resolution.

| Residual connection | ✗ | ✔ | ✔ |
|---|---|---|---|
| Extra key-value | ✗ | ✗ | ✔ |
| PSNR ↑ | 23.253 | 23.483 | **23.582** |
| $\mathcal{L}_1$ ↓ | 12.741 | 12.721 | **12.632** |
| LPIPS ↓ | 0.122 | 0.114 | **0.113** |
| AKD (MTCNN) ↓ | 1.969 | 1.914 | **1.895** |
| FID ↓ | 19.978 | 18.462 | **18.252** |

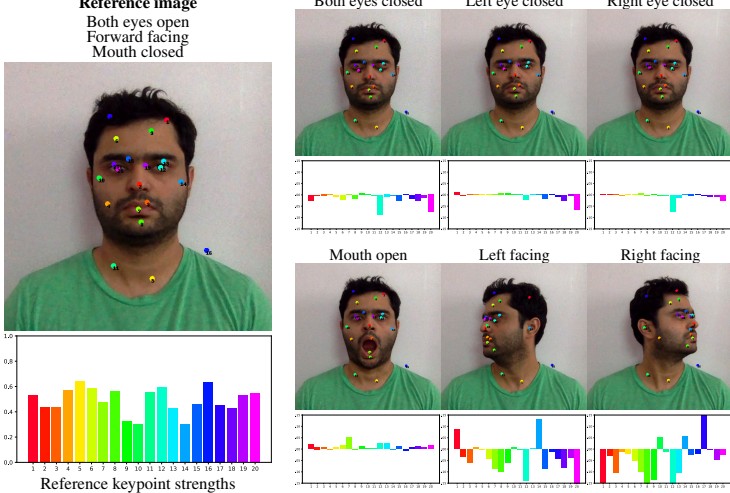

Fig. 6: On the left, we show a reference image overlayed with predicted keypoints, and their strengths. On the right, we show different input images and their relative keypoint strengths w.r.t. the reference keypoint strengths (Please zoom in to see numbered keypoints on the face). The network latently learns to associate different values of the strength to different keypoint visibilities and facial configurations.

the correct frontal face. Since it is a data-driven approach, it might fail for extreme expressions not present in the training dataset. Additional data and augmentations might help alleviate such issues.

**Societal impact.** Our method has the potential for negative impact if used to create *deepfakes*. Via the use of cross-identity transfer and speech synthesis, a malicious actor can create faked videos of a person, resulting in identity theft or dissemination of fake news. However, in controlled settings, the same technology can also be used for entertainment purposes. Our method can be used in low-bandwidth video compression in the same-identity video reconstruction setting. This can help improve video quality, especially for regions with poor connectivity. As our method generates outputs using source image features, we have not observed and do not expect it to demonstrate any racial bias.

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
