# OpenReview forum: "Implicit Warping for Animation with Image Sets"
_NeurIPS.cc/2022/Conference — NeurIPS 2022 Accept_

### Official Review · Reviewer_BA5v · 2022-07-07

**Rating:** 7
**Confidence:** 4
**Soundness:** 3 good
**Presentation:** 3 good
**Contribution:** 3 good

**Summary:**

The paper presents a method for warping-based image animation using multiple source images (i.e., images of the scene to be animated) and a driving video (i.e., image frames from the same or similar scene used to drive the animation). Since previous methods have focused mostly on animating scenes with a single source image, including multiple source images in these frameworks is non-trivial. Yet, it makes sense that multiple source images could improve performance since they may provide textures from occluded regions that can be used in the animation. The key technical contribution is to combine existing keypoint-detection methods with an attention-based warping. This allows the model to attend to image features across all input frames in order to produce a dense grid of warped output features. The method is evaluated for face and body animation and demonstrates state-of-the-art results for both single and multiple input source images.


**Questions:**

- The main paper doesn't seem to describe how the outputs of the cross-modal attention are upsampled to the final output resolution. While this is clear from the supplement, I think a sentence should be added for completeness.

- I wasn't entirely sure what the keypoint representation is. In the FOMM paper, the keypoint predictor produces a heatmap that is passed through a softmax to produce a probability distribution whose mean is used to calculate a pixel location. Here, the keypoints seem to be passed directly into a convolutional layer so are the heatmaps (pre- or post-softmax) used for the keypoint representation?

- How much does the number of keypoints matter in this case? I see that 20 keypoints are used for all experiments, but I'm curious how much this hyperparameter matters.

- How are the extra/added key-value pairs implemented? Are these simply learnable parameter vectors of dimension d? Or are they conditioned somehow on the input source images?

- I was curious why the FOMM model was selected for baseline comparisons rather than the more recent work from the same authors which addresses some of the shortcomings of the original model (Siarohin et al., ref. below). If I look at the quantitative results from that paper, they seem close to the proposed method, e.g., for AKD on the TED Talk dataset. Is there a reason this baseline wasn't included?

    - Siarohin, Aliaksandr, et al. "Motion representations for articulated animation." Proc. CVPR. 2021.


**Limitations:**

- The paper briefly touches on computational efficiency, but I think this limitation could be clarified a bit more. How does the proposed attention-based model compare to the baseline methods in terms of efficiency? I would guess that FOMM or FV2V have a much higher framerate, and the proposed model achieves better quality, but with some penalty to efficiency. Additional clarification on this point would be helpful.

- There are obvious ethical concerns with the proposed method, especially with respect to potential misuse for deep fakes. Still, I think the authors address this point reasonably well in the societal impact section while highlighting positive benefits of this class of technique.

Typos
Fig. 3: "Coss-model" -> "Cross-modal"


**Strengths And Weaknesses:**

*Strenghts*

- The paper is well-written and the technical descriptions are clear and fairly easy to follow.

- This method seems to elegantly and sensibly address the challenge of warping multiple input frames into the single output using information from the driving video. The implicit warping formulation using the attention mechanism helps to solve some limitations of previous methods (which could only support explicit, linear warping). I could see this approach being relevant to other problems in image-based rendering or sensor fusion.

- The method clearly outperforms the other baselines and the additional results (included in the videos and webpage) are compelling.

*Weaknesses*

- While the paper is generally easy to follow, there are some technical details missing from the main paper (see questions below).

- Some aspects of the related work section could be improved. For example the "Image Animation" section notes that "none of [these frameworks] is designed to take advantage of complementary information available in different input images." Yet, there are a number of papers that use multiple input images, e.g., for facial animation [30-33]. In these works, multiple input images can be used to synthesize a dense texture map of the face, which can then be animated using a 3DMM style model. Some rewording here could help make this section more accurate.

- One other limitation of the method seems to be scalability. While there are results demonstrated on 512x512 images, achieving higher resolution seems difficult since the attention-based mechanism happens at a relatively low resolution (the 64x64 resolution is mentioned in the paper). The features predicted after the attention module are then upsampled afterwards to produce the a higher resolution output, but I suspect there's a limit to how much upsampling can be done while retaining good quality.

Overall I think these are relatively minor weaknesses; the method appears sufficiently novel and original, and the results seem quite strong.

---

> ### Author Response · Authors · 2022-08-01
> **Response to Reviewer BA5v**
>
> **Resolution of cross-modal attention:** As mentioned in Lines 149-150, the keys, queries, and values are produced at 1/4th resolution of the input images, i.e. 64x64 for 256x256, and 128x128 for 512x512 input images. The corresponding attention maps and warped features are also at 64x64, and 128x128 resolutions respectively. Our decoder network has 2 upsampling layers that produce an output image at 4x the resolution of the warped features, similar to FOMM and face-vid2vid, as shown in Fig. 9 in the supplementary material. For faces, we first train at 256x256, and then finetune at 512x512, without any changes to the network architecture. We will make this clear in the main text.
>
> **Keypoint representation and its usage:** Similar to FOMM, we use a keypoint predictor network to predict keypoint locations by applying a spatial softmax on its outputs. Additionally, we also predict a scalar per-keypoint in the range [0-1]. Similar to FOMM, in the decoder, we place a Gaussian of fixed mean and variance at the predicted keypoint location. We further modulate these spatial keypoint representations by multiplying them with the associated scalar. These modulated features of *K* channels (where *K* is the number of keypoints) are fed as input to the query and key networks, as shown in Fig. 9.
>
> **Effect of number of keypoints:** We used 20 keypoints in all comparisons following the prior work of face-vid2vid. We found that increasing the number of keypoints from 20 to 30 on the TalkingHead-1KH dataset improved the quality of results — increasing PSNR from 23.32 to 24.26, reducing AKD (MTCNN) from 3.48 to 3.18, and L1 from 12.86 to 11.92, showing improved reconstruction and keypoint fidelity.
>
> **Newer baselines:** We actually do use the newer work "Motion representations for articulated animation." Proc. CVPR. 2021, in our comparisons on the upper-body TED-Talk dataset in Table 1, and Fig. 5. Note that this method is referred to as AA-PCA [28] (Articulated Animation using PCA), also noted in Lines 208-209. For faces, we compare with the state-of-the-art face-vid2vid. We will make this naming convention clear.
>
> **Method efficiency comparison:** When using a single source image at 512x512 output resolution, the FPS of FOMM, face-vid2vid, and implicit warping (ours) are 13.3, 7.1, and 9.6. When using 3 source images, the corresponding FPS are 9.5, 3.6, and 5.1. Our method is slower than FOMM but faster than face-vid2vid. These framerates were obtained on an NVIDIA RTX A6000 GPU.
>
> **Extra key-and-values implementation:** Please see common reply comment for detailed answer.

---

### Official Review · Reviewer_U2o1 · 2022-07-13

**Rating:** 7
**Confidence:** 3
**Soundness:** 3 good
**Presentation:** 3 good
**Contribution:** 3 good

**Summary:**

This paper presents a novel method for transferring the motion of a driving video to a subject via an image set of the subject. By using a cross-model attention layer for directly generating warped features, the proposed method differs from most previous methods which find explicit correspondences and warp images accordingly. By using the proposed mechanism, the proposed method can benefit from multiple source images without resorting to ad hoc approaches and suffering from sub-optimal results such as flicker or blur.

**Questions:**

Could you please provide more details regarding the procedure for adding additional keys and values?

Is there a reason why the ablation study was conducted at a reduced resolution?


**Limitations:**

The paper discusses the societal impacts adequately. The method may fail if it is required to fill in a large amount of missing information. Additionally, it may encounter problems when dealing with extreme expressions not present in the training data.

**Strengths And Weaknesses:**

Overall, the paper appears to be very promising. The paper describes a simple yet effective method of generating an animation from a set of images. In the case of multiple images, the proposed method clearly outperforms other methods. Though the proposed attention layer is commonly used in many applications, its use in this problem setting and the overall architecture design are novel.

### Strengths ###
* The proposed method is both simple and effective. The core of the algorithm is the cross-model attention layer. Although the idea has been used extensively in many applications, its use in this setting is novel to my knowledge.

* The results of the experiments are promising and convincing. The proposed method was compared with a couple of SOTA methods using multiple datasets. A major advantage of the proposed method is its ability to utilize multiple images simultaneously and effectively.

* The paper is well written and enjoyable to read. The supplementary material, in particular the video, is very well done.

### Weaknesses ###
* The cross-modal attention layer does not present much technical innovation. However, the overall flow and architecture design are novel.

* It is not entirely clear how to add additional keys and values. It would be helpful if a more detailed description was provided.

* It is not clear why the ablation study was conducted at a reduced resolution.

---

> ### Author Response · Authors · 2022-08-01
> **Response to Reviewer U2o1**
>
> **Ablation at reduced resolution:** Our networks on the TalkingHead-1KH dataset are trained in two steps, as detailed in Section D of the supplementary material — first at 256x256 resolution, followed by finetuning at 512x512. The first stage takes 3.5 days (7 days) on an NVIDIA DGX server with 8 A100 40GB GPUs (8 V100 32GB GPUs). The second stage takes about a day on the DGX A100, or 2+ days on the DGX V100. Due to the limited computational resources, combined with experiments on multiple datasets and settings, we chose to perform ablations at the 256x256 resolution. Each ablation experiment was run 3 times. We then chose the best networks and finetuned them at 512x512 for faces, and trained them at 384x384 from scratch for the upper body TED-Talk dataset.
>
> **Extra key-and-values implementation:** Please see common reply comment for detailed answer.

---

### Official Review · Reviewer_hL38 · 2022-07-22

**Rating:** 7
**Confidence:** 3
**Soundness:** 3 good
**Presentation:** 3 good
**Contribution:** 2 fair

**Summary:**

This paper concerns the task of animating a set of one or more source images driven by a target (or driving) video; applications include efficient video compression (e.g. for video calls) or retargeting a video.

Out of a four stage pipeline, the authors note that their work focuses on the warping of image features from the source image(s) to the output. Previous work provides feature representation extraction from the source and driving images, we well as decoding from from the warped feature representation to the final output image.

The main contribution is the direct application of a dot-product attention based transformer to perform implicit warping between the driving and source image(s); this is an elegant solution that requires no special treatment for different numbers of source images. This is contrasted against previous approaches that only consider a single source image without an obvious method to extend to multiple sources images that avoids undesirable image operations (e.g. averaging over output images or hard transitions between source images).








**Questions:**

Additional keys: The use of the additional keys seems relatively critical to the method (particularly some of the outputs where occlusion means there are no suitable source regions to use). The ablation study for this is relatively simple (as a binary consideration) whereas there are a number of parameters for the additional keys. At the same time, the ablation study results are not very conclusive (Table 5 in the supplement); the improvements for the majoritiy of the metrics are well within the associated standard error values provided. It would seem that a further investigation of this area is waranted - as the authors point out, it should be vital to the success but this doesn't seem to follow through to the ablation study metrics - please could the authors comment on this? Is there not a need for further study here and how were the parameters (e.g. the number of additional keys) derived? It would also seem that these additional keys become very application specific (e.g. synthesising eyes); does this limit the general applicability of the results across datasets?

Error bars/Variance: Given the contrast between Table 4 and Table 5 (where std errors are provided) it seems necessary to provide the std errors to the other results tables - it is difficult to judge the significance of the improvement and we would suspect, from the Table 5 std dev, that these improvements may not be particularly significant given the dataset (e.g. table 1 and 2). Please can these values be provided and the authors comment on this - if this is not important, please could the authors indicate why it should be neglected in the analysis of the quantitative results?

Mismatch in the results: Please could the authors explain why the first columns of Table 2 don't match the corresponding columns in Table 1 (i.e. the results for single source images)? Sorry if I have missed something.

Application Level Evaluation: Please could the authors comment on the intended application for such work? For example, is the goal retargetting or also efficiency for (e.g.) video calls? It seems the information required per frame (the A matrix) exceeds the original image dramatically therefore there would not be a reduction in bandwidth if the result is to be sent over a communications channel? How would this compare with the need to send the warping field (which might be readily compressible)? The authors specifically cite this in lines 257-258 as an application and it seems to me that no data compression has been achieved? Or is it the case that we don't need to send q x d and that the driving keypoint network is fixed so only the keypoint input data can be sent?

**Limitations:**

The authors flag that operating under high levels of occlusion will be very challenging (this makes sense and the arguments that multiple images help with this is reasonable - perhaps a consideration of how the source images should be chosen would be helpful)? There is also discussion that computational/memory complexity could be improved (under the observation that there is perhaps significant sparsity that has not been exploited).

Comments that might want to be considered by the authors but I don't believe warrant ethical review:

In consideration of societal factors, the authors point out that the final video is generated from the source image and therefore they would not expect racial bias in the results; whilst this makes sense for one part, I think this misses the consideration that the method relies on the key point, representation networks and feature decoders to operate well and these are all trained on data - if these datasets contain significant bias would we not expect the performance to vary as a result?

The authors included the evaluation protocol and payment for the human evaluation study; I noted that the estimated hourly rate provided ($5/hour) is demonstrably lower than the US minimum wage (around $7.5/hour).


**Strengths And Weaknesses:**


Whilst the direct application of a transformer, the resulting approach is an elegant solution that prioritises simplicity for which I commend the authors. I found the presentation of the method to be straight-forward and clear - I believe the use of the different components is logical and well justified.

The proposed method offers a number of advantageous properties over previous approaches whilst removing complexity. I believe it is fair to say that the empirical evaluation should be of critical consideration in determining the merits of the approach as it determines whether or not the the hand specified architectural and modelling decisions (or implicit priors) in previous approaches can be relaxed in favour of a data driven approach. For this reason, this is the main focus of the review.

In terms of the quality of the results, I have a number of questions regarding the quantitative evaluation. This also speaks to the significance of the results in terms of how well we expect the method to generalise and how well it would perform at downstream tasks. There are a number of points that I currently see to be weaknesses in the evaluation; please see the questions below for my comments on this which include specific questions for the authors to clarify.

Caveat: This is not my main area of research; whilst I am familiar with the methods and concepts being brought to bear, I am not familiar with the literature in the application area and cannot speak to the novelty or otherwise for the application; I cannot guarantee that there is not related work that I have missed.

---

> ### Author Response · Authors · 2022-08-01
> **Response to Reviewer hL38**
>
> **Error bars and multiple runs:** Our networks on the TalkingHead-1KH dataset are trained in two steps, as detailed in Section D of the supplementary material — first at 256x256 resolution, followed by finetuning at 512x512. The first stage takes 3.5 days (7 days) on an NVIDIA DGX server with 8 A100 40GB GPUs (8 V100 32GB GPUs). The second stage takes about a day on the DGX A100, or 2+ days on the DGX V100. Due to the limited computational resources, combined with experiments on multiple datasets and settings, we chose to perform ablations only at the 256x256 resolution. Each ablation experiment was run 3 times. We then chose the best networks and finetuned them at 512x512 for faces. The best settings found were also used for the voxceleb2 and TED-Talk datasets.
>
> The ablations in Table 5 do demonstrate high variance, but we would like to note that the best metrics over all runs were obtained by the setting in the last column of Table 5. Below, we provide the best metrics obtained over 3 runs for the columns of Table 5:
> | Metric | No residual or extra key-value | Residual connection only | Residual connection + Extra key-value |
> | :---        |    :----:   |    :----:   |    :----:   |
> | FID      | 19.10 | 18.32 | **17.53** |
> | PSNR  | 23.49 | 23.63 | **23.82** |
> | LPIPS | 0.117 | 0.112 | **0.109** |
> | AKD (MTCNN) | 1.953 | 1.903 | **1.827** |
>
> Based on this observation, we chose to use the last column setting of residual connection +  extra keys and values, for our main experiments. We will provide detailed metrics for each of the 3 runs in the supplementary material.
>
> **Additional keys become very application specific:** Yes, this is true. The trained model, including the learned keys and values are not transferable across very different datasets, for e.g. from faces to half-body videos. Details about the implementation are available in the common comment section.
>
> **Mismatch in Table 1 and Table 2:** Table 1 measures metrics for image reconstruction while using a single source image, while Table 2 measures metrics for image reconstruction using multiple source images. As mentioned in Line 231, for Table 2, we use at most 180 frames per video during evaluation. For Table 1, we used the entire videos, which have up to 1024 frames (Line 477 in supplementary). Hence the difference. We will clarify this in the updated draft.
>
> **Application-level evaluation:** Our proposed method can be used for both motion retargeting, as well as video reconstruction for conferencing purposes. In the case of video conferencing, the sender’s side (encoder) only has the keypoint detector network shown in Fig. 8. The receiver’s side (decoder) contains the query, key, and value networks as shown in Fig. 9.
> Note that we do not have to transmit the q x d sized attention matrix A — we only have to transmit the keypoint locations and associated scalar keypoint strengths. After training, all networks are fixed and the attention matrix can be obtained on the decoder side using the keypoint location and strength information as inputs to the query, key, and value networks.
>
> **Limitations of keypoint detector:** This is an excellent observation! If the keypoint detector is trained on a biased dataset, it might not perform well for all skin colors, lighting conditions, etc. We will emphasize that training on a diverse dataset is a must, to ensure wide coverage of the keypoint detector as well as the decoder stack.

---

> > ### Comment · Reviewer_hL38 · 2022-08-07
> > **Thanks for the response - updated review to Accept**
> >
> > I thank the authors for their detailed response and my main concerns are addressed. I would encourage the authors to make the updates they describe and am happy to upgrade my review to Accept.

---

### Author Response · Authors · 2022-08-01
**Response to all reviewers on common questions/comments**

**Extra key-and-values implementation:**
In our implementation, the extra keys and values are learned by the network during training. We initialize new parameters in the decoder, corresponding to the extra keys and values, as shown in the Python PyTorch code snippets below:
```python
# Initialization.
​​self.extra_k = nn.Parameter(torch.randn(1, num_extra_kv, dim_qk))
self.extra_v = nn.Parameter(torch.randn(1, num_extra_kv, dim_v))
self.norm_extra_k = nn.LayerNorm(dim_qk)
self.norm_extra_v = nn.LayerNorm(dim_v)
```


During the forward pass, these extra keys and values are appended to the keys and values obtained from the input source image(s):
```python
# Forward pass.
# k, v are from the source image(s).
Bk, Nk, Ck = k.shape
Bv, Nv, Cv = v.shape
k = torch.cat((k, self.norm_extra_k(self.extra_k).expand(Bk, *self.extra_k.shape[1:])), dim=1)
v = torch.cat((v, self.norm_extra_v(self.extra_v).expand(Bv, *self.extra_v.shape[1:])), dim=1)
```

After training, the above extra keys and values are kept fixed.
These extra keys and values are not conditioned on the input source or driving images. They are static learned vectors that are specific to the dataset and task, similar to the weights of the rest of the network. We can also interpret them as a learned dictionary mapping.

---

### Meta-Review · Area_Chair_6pKa · 2022-08-27

**Recommendation:** Accept
**Confidence:** Certain

**Metareview:**

Consistent reviews, both in content and in score.

The cross-identity motion transfer is a good test of the paper's capability -- it would improve the paper to provide more such examples, which are clearly more challenging than the same-identity case.

The concerns about the limited diversity of example subjects mentioned by R1 are indeed relevant.  The video examples are all male, with quite light skin tone.  Please include examples with female subjects, darker (Fitzpatrick 6+) skin tone, and other ethnicities.  To be clear: the rebuttal's current response "we will emphasize that training on a diverse dataset is a must" does not go far enough.  It is very important that qualitiative examples are shown, even more important than that the test datasets are diverse.  If the results are less good, efforts should be made before NeurIPS to improve them (e.g. by retraining), and if improvement is not possible, this should be very clearly stated in the limitations of the final copy and NeurIPS presentation/poster.


**Award:**

No

---

### Decision · Program_Chairs · 2022-09-14

Accept